# Gait speed and incline modulate peak deceleration and timing of horizontal center of mass deceleration during double support

Shizuku Terui[1], Nanami Kanda[2] and Keisuke Hirata[1,2,‡]

## ABSTRACT

Adapting gait to varying speeds and inclines is essential for navigating complex environments. The movement of the center of mass (CoM) in the horizontal plane during the double-support phase is considered critical for maintaining gait performance, but how specific CoM deceleration patterns adapt to these challenges is not fully understood. This study examined how gait speed and incline affect the most deceleration (MD) and its timing (MDt) of CoM movement in the horizontal plane during the double-support phase of gait in healthy individuals. Fourteen healthy young adults walked on a treadmill under four conditions combining two speeds (moderate: 0.83 m/s, fast: 1.0 m/s) and two inclines (level: 0°, uphill: +6°). CoM movements were recorded using a motion capture system. Key parameters analyzed included double support time ratio (DST), step length (SL), MD and MDt. SL increased with speed but was not significantly affected by incline at matched speeds; DST remained unchanged across conditions. Crucially, MD significantly increased with both faster speed and incline, being largest under the uphill-fast condition. Furthermore, MDt occurred significantly earlier in the gait cycle during faster and uphill conditions compared to moderate-speed level walking. This peak deceleration consistently occurred just prior to contralateral toe-off. Our study concludes that healthy young adults adapt to increased gait speed and incline by modulating both the deceleration and timing of CoM movement in the horizontal plane during double support. The increased deceleration and its earlier timing, particularly under challenging conditions, may reflect kinematic adaptations related to momentum regulation and step-to-step coordination, rather than indicating of neuromuscular control. These findings provide insight into potential mechanisms underlying gait adaptation in healthy individuals.

KEY WORDS: Gait speed, Incline, Double support phase, Gait, Centre of mass

## INTRODUCTION

The double-support phase of gait involves a smooth weight transition from the trailing leg to the leading leg. This transition is crucial for efficient forward progression, minimizing energy loss, and mitigating abnormal loads potentially caused by ground

reaction forces (Adamczyk and Kuo, 2009). Individuals with unilateral lower limb impairments, such as anterior cruciate ligament reconstruction patients (Grapar Zargi et al., 2017), unilateral amputees (Norvell et al., 2005), and stroke hemiplegic patients (Little et al., 2018) often experience difficulties in weight transition. While this phase is fundamental for stable locomotion in healthy individuals, the specific control mechanisms employed by the central nervous system, especially concerning how the body's center of mass (CoM) dynamics adapt to common challenges like varied gait speeds and inclines, require further elucidation.

Gait speed and incline are known to influence the spatiotemporal characteristics and mechanics of the double-support phase (Dewolf et al., 2017). Understanding how CoM motion is controlled during this phase under such varying conditions is essential. In particular, the control of CoM velocity in the horizontal plane (encompassing mediolateral and anteroposterior directions) is considered critical for maintaining dynamic gait stability (Vlutters et al., 2016). Fluctuations in horizontal CoM velocity are thought to reflect underlying balance control strategies, and deficits in this control may be associated with an increased risk of falls (Vlutters et al., 2016), underscoring the importance of analyzing dynamics within this plane.

While the overall CoM trajectory provides valuable information, the deceleration of the CoM during the late double-support phase, specifically around the time of the trailing leg's toe-off, likely plays a pivotal role. This deceleration event is critical for modulating forward momentum and ensuring a smooth, stable transition into the subsequent single-support phase. Previous studies have analyzed aspects of CoM mechanics and energy during gait transitions and varied conditions. Adamczyk and Kuo (2009) focused on the redirection of CoM velocity and the work performed on the CoM during the step-to-step transition, comparing human behavior to rigid-leg pendulum models. They showed that the angular redirection of CoM velocity and the work performed tend to increase with walking speed and step length. However, they noted that their simple model could not predict the duration of the step-to-step transition, which empirically exceeds the double support period. Dewolf et al. (2017) analyzed within-step energy conversion (exchange between kinetic and potential energy) and external work during walking across different slopes and speeds (Dewolf et al., 2017). They demonstrated that slope and speed affect these energy-related parameters and introduced a new variable (pendular energy savings, Es) to quantify the effectiveness or magnitude of energy exchanged within the step. Unlike Adamczyk and Kuo (2009) who explored predictors of work, Dewolf et al. (2017) primarily focused on the energy conversion efficiency and total external work (Dewolf et al., 2017).

While these foundational studies have provided valuable insights into overall CoM mechanics, transition work, and within-step energy transformations, they did not specifically investigate or quantify the most deceleration and its timing of CoM movement in the horizontal plane, particularly within the double-support phase,

[1]Department of Rehabilitation, Faculty of Health Sciences, Tokyo Kasei University, 2-15-1 Inariyama, Sayama-shi, Saitama 350-1398, Japan. [2]Graduate Course of Health and Social Services, Graduate School of Saitama Prefectural University, 820 Sannomiya, Koshigaya-shi, Saitama 343-8540, Japan.

‡Author for correspondence (hirata-ke@tokyo-kasei.ac.jp)

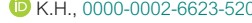 K.H., 0000-0002-6623-5203

Biology Open

as key kinematic variables reflecting momentum management and adaptation to varying speed and incline. We contend that the maximum deceleration of the CoM in the horizontal plane and its timing within the gait cycle, particularly during the double support phase near trailing leg toe-off, represent critical kinematic variables that reflect how the body manages its momentum transfer between steps. Investigating how these specific parameters adapt to changes in speed and incline is therefore essential for gaining deeper insights into the adaptive neuromuscular strategies governing walking.

Therefore, the primary purpose of this study was to clarify the effects of gait speed and incline on these specific characteristics – the most deceleration and its timing of CoM dynamics in the horizontal plane during the double-support phase in healthy young adults walking on a treadmill.

We hypothesized the following: (1) both increased gait speed and the addition of an incline would increase the most deceleration (MD) and its timing (MDt) of CoM movement in the horizontal plane during the double-support phase, and (2) these task modifications would shift MDt to occur earlier within the gait cycle, potentially indicating anticipatory kinematic adjustments to accommodate altered biomechanical demands.

Understanding these kinematic responses during healthy gait provides an essential baseline for comparison and may ultimately inform our understanding of gait impairments. Although this study focuses exclusively on healthy young individuals, the findings regarding how CoM deceleration patterns are modulated may offer insights into movement adaptations that are relevant to maintaining gait function under varying task constraints. Such knowledge could be useful in informing the design of interventions aimed at enhancing movement efficiency or safety, for example, in fall prevention or rehabilitation contexts.

## RESULTS

All variables passed the normality test using the Lilliefors test ($P>0.05$, Table 1). The exploratory subgroup analysis revealed no significant sex differences in MD, MDt, SL, or DST under any of the four conditions ($P=0.09-0.93$).

The DST showed little variation across all conditions (7.56–9.88±1.92–2.05%), and no significant differences were observed between the conditions [$F_{(3, 42)}=1.23$, $P=0.31-0.98$, Fig. 2A].

Significant differences in SL were found between level-moderate and level-fast, uphill-moderate and uphill-fast, and level-moderate and uphill-fast conditions ($F_{(3, 42)}=6.32$, $P=0.003-0.02$, Fig. 2B]. Step length was significantly greater in the faster speed conditions. However, no significant differences were observed between level-moderate and uphill-moderate ($P=0.35$) or between level-fast and uphill-fast conditions ($P=0.42$, Fig. 2B).

The MD showed significant differences between uphill-fast and the other conditions (level-moderate, level-fast, and uphill-moderate), with uphill-fast exhibiting the highest deceleration ($-1.77±0.61$ m/s³,

$P=0.004-0.03$). No significant differences were found between level-moderate and level-fast, or between level-moderate and uphill-moderate ($P=0.25-0.47$).

The MDt differed significantly between level-moderate and the other conditions (level-fast, uphill-moderate, and uphill-fast) [$F_{(3, 42)}=7.85$, $P=0.0005-0.01$]. The faster speed and uphill conditions showed earlier maximum deceleration (46–49%) compared to level-moderate (53.48±0.97%) (Fig. 3).

The difference between the MDt and the iHC was 3.22–7.72% across all conditions (mean difference in Bland-Altman analysis: 5.11–72) (Fig. 4A). In contrast, the difference from the cTO was 1.23–2.89% across all conditions (mean difference in Bland-Altman analysis: $-2.89$ to $-1.23$), indicating that the cTO was significantly closer to the maximum deceleration point compared to the contralateral heel contact point. Since the mean difference was negative, it was found that the maximum deceleration was observed just before cTO (Fig. 4B).

## DISCUSSION

This study aimed to clarify the effects of gait speed and incline on the MD and MDt of peak CoM deceleration in the horizontal plane during the double-support phase in healthy young adults. Our primary findings supported our hypotheses: both faster speeds and uphill walking significantly increased MD, and these conditions shifted the MDt earlier in the gait cycle compared to moderate-speed level walking. These results indicate that alterations in CoM deceleration patterns near toe-off are characteristic of how gait adjusts to changing mechanical demands.

The observed increase in MD, particularly under the challenging uphill-fast condition (Fig. 2C), suggests a greater requirement to manage the body's momentum during the critical transition between steps. This larger deceleration likely reflects the increased mechanical demands imposed by higher speed and the need to counteract greater gravitational forces during uphill walking. Concurrently, the earlier occurrence of MDt under faster and inclined conditions (Fig. 3) may reflect anticipatory kinematic adjustments that help facilitate step-to-step transition. Initiating deceleration earlier relative to the subsequent heel contact could represent a biomechanical adaptation that ensures adequate time and body configuration for weight transfer onto the leading limb, especially under demanding conditions (Hak et al., 2012; Martino et al., 2015). While we did not directly quantify gait stability or neural control, the systematic modulation of MD and MDt observed here could be interpreted as part of a broader pattern of movement adaptation during complex walking tasks. However, it is crucial to acknowledge that changes in incline directly alter the gravitational forces acting on the body, intrinsically influencing CoM dynamics. Therefore, the observed kinematic changes likely result from an interplay between these altered environmental demands and the neuromuscular system's adaptive responses, rather than providing direct evidence of a specific control mechanism. Further research incorporating kinetic and electromyographical analyses is warranted to more definitively disentangle these contributing factors.

Interestingly, our findings regarding spatiotemporal parameters diverge somewhat from previous studies. While SL predictably increased with speed, it did not significantly change between level and uphill conditions at the same speed (Fig. 2B). Furthermore, the DST remained remarkably consistent across all tested conditions (Fig. 2A). Previous research comparing level and uphill gait, often employing protocols where walking speed was reduced on inclines to approximate similar metabolic costs, typically reported shorter step lengths and longer double support times during uphill walking

## Table 1. The means and standard deviations (s.d.) of the maximum deceleration of the CoM on the horizontal plane defined in the anteroposterior and MD, and its timing (MDt, maximum deceleration point normalized to 100% of the gait cycle)

| | MD (m/s²) | | | MDt (%) | | |
|---|---|---|---|---|---|---|
| | Mean | | s.d. | Mean | | s.d. |
| Level-moderate | 3.26 | ± | 1.08 | 53.49 | ± | 0.98 |
| Level-fast | 3.28 | ± | 1.23 | 49.17 | ± | 0.98 |
| Uphill-moderate | 3.55 | ± | 1.45 | 46.52 | ± | 1.01 |
| Uphill-fast | 5.06 | ± | 1.63 | 47.09 | ± | 1.05 |

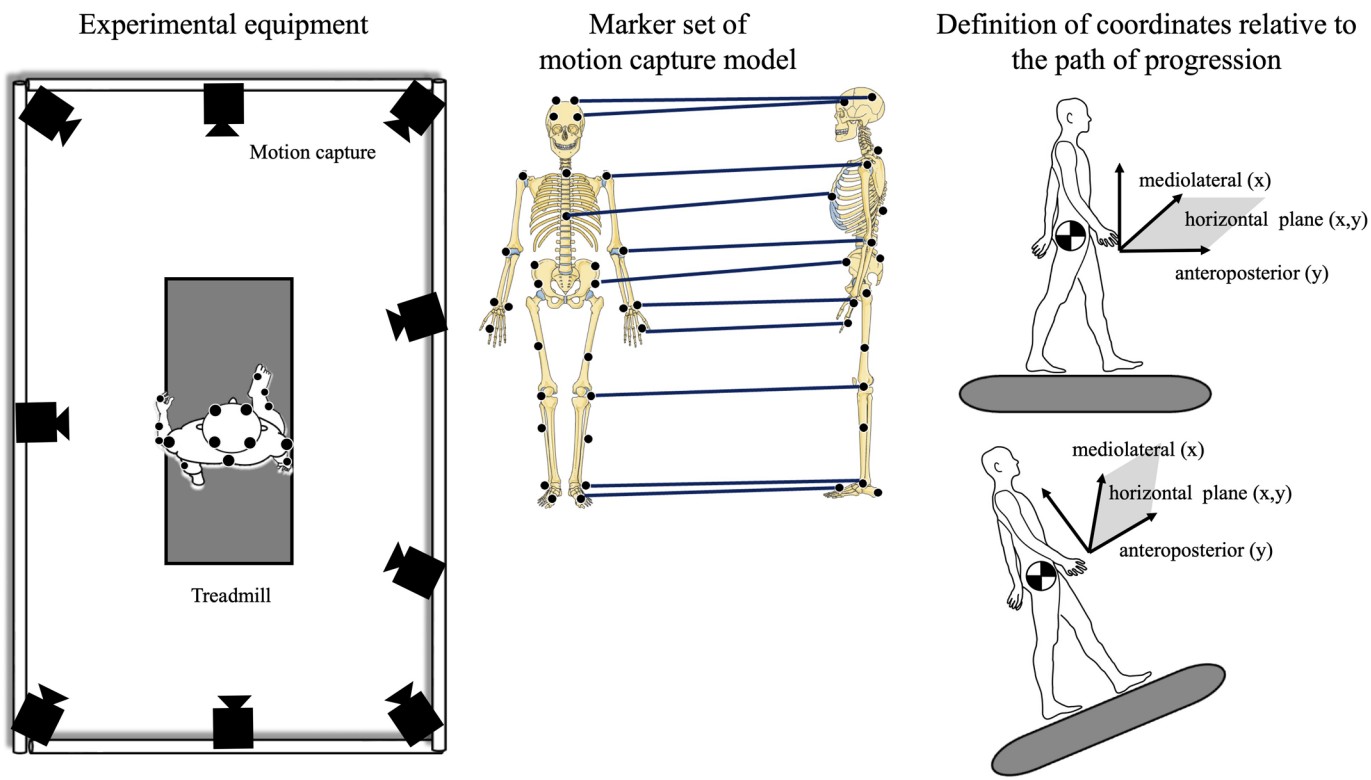

**Fig. 1. Participants walked on a treadmill equipped with an incline function under conditions combining speed and incline, and their motion was captured using a motion capture system to obtain the coordinates of the CoM.** For the uphill conditions, the anteroposterior coordinates were transformed using trigonometric functions to align with the gait path, similar to the level conditions. Therefore, both the anteroposterior direction and the horizontal plane defined by the anteroposterior and mediolateral directions were made parallel to the gait path in both level and uphill conditions. Parts of the figure were drawn by using pictures from Servier Medical Art. Servier Medical Art by Servier is licensed under a Creative Commons Attribution 3.0 Unported License (https://creativecommons.org/licenses/by/3.0/).

(Kafetzakis et al., 2024; Milic et al., 2020). In contrast, our study maintained consistent walking speeds across level and uphill conditions. The absence of significant changes in SL and DST between level and uphill gait in our study suggests that participants prioritized maintaining the imposed locomotor rhythm. This was likely achieved by increasing net mechanical work and muscular effort (indirectly reflected by the increased MD) rather than altering these fundamental spatiotemporal parameters. This underscores how specific task constraints (e.g. fixed speed versus fixed effort) significantly shape gait adaptations. The implications of these adjustments for postural control remain speculative, and future work

incorporating direct stability-related metrics would be necessary to address this explicitly.

Our analysis of the timing of the peak deceleration event relative to gait cycle landmarks provides further insight. The Bland–Altman plots revealed that MDt consistently occurred just prior to the cTO across all conditions (Fig. 4B), demonstrating a tight temporal coupling. This precise timing may indicate that the peak horizontal deceleration is kinematically coordinated with the termination of the trailing limb's propulsive phase. Such timing could be biomechanically advantageous for modulating momentum and preparing for swing initiation.

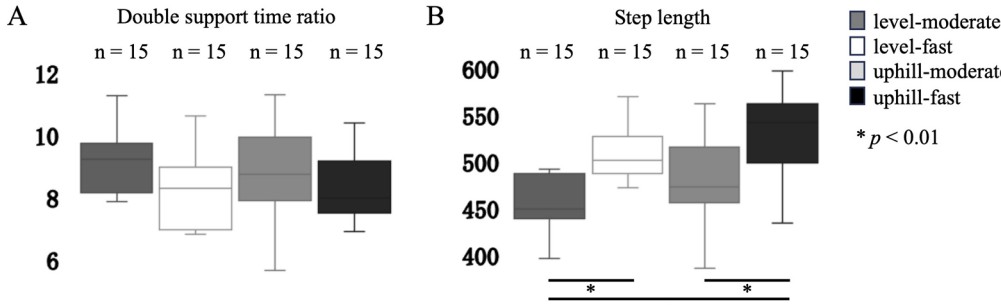

**Fig. 2. The double support time ratio showed little variation across all conditions (7.56–9.88±1.92–2.05%), and no significant differences were observed between the conditions [$F_{(3, 42)}=1.23$, $P=0.31–0.98$, A].** Significant differences in step length were found between level-moderate and level-fast, uphill-moderate and uphill-fast, and level-moderate and uphill-fast conditions [$F_{(3, 42)}=6.32$, $P=0.003–0.02$, B]. Step length was significantly greater in the faster speed conditions. However, no significant differences were observed between level-moderate and uphill-moderate ($P=0.35$) or between level-fast and uphill-fast conditions ($P=0.42$, B).

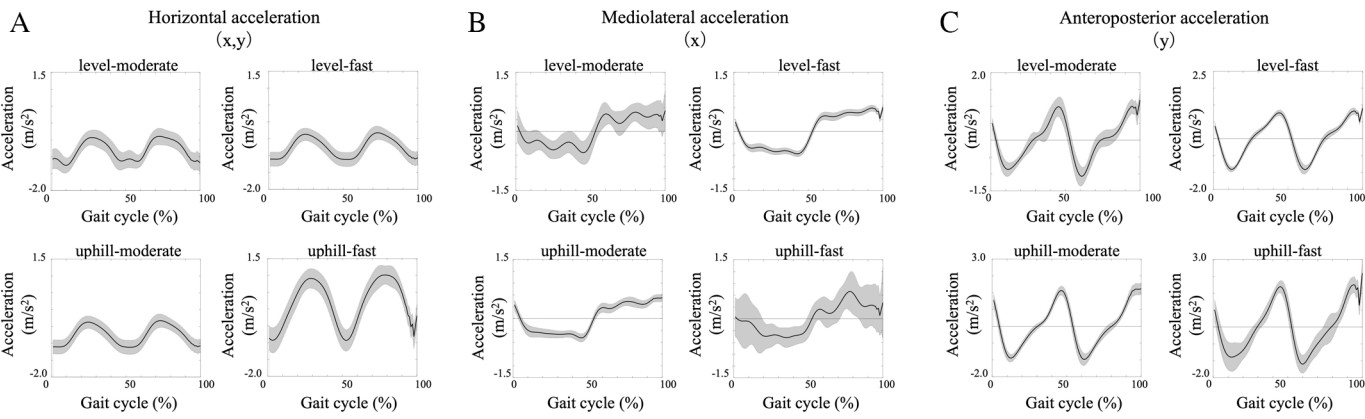

**Fig. 3. An example of velocity (A) and acceleration (B) in the horizontal plane (left column), mediolateral direction (center column), and anteroposterior direction (right column), normalized to 100% of the gait cycle (from right heel contact to the next right heel contact).** The Y-axis represents: positive values for the horizontal plane indicate rightward and forward movement, and negative values indicate leftward and backward movement; in the mediolateral direction, positive values indicate rightward and negative values indicate leftward movement; in the anteroposterior direction, positive values indicate forward movement and negative values indicate backward movement. In the top row, velocity traces for all steps from one example are overlaid, while in the bottom row, the average acceleration for all steps of one example is shown in black, with the standard deviation in gray.

Furthermore, the results indicated a pronounced CoM deceleration in the horizontal plane near toe-off, which appeared directed backward and toward the stance (trailing) leg. While some deceleration is mechanically expected as propulsive forces diminish and gravity acts (especially on an incline), the significant modulation of the MD and MDt by speed and incline suggests more than just a passive mechanical consequence. The earlier MDt under challenging conditions, coupled with the increased MD, may reflect an adaptive response aimed at regulating momentum transfer between limbs. For instance, the well-documented increase in ankle plantar flexor activity (soleus and gastrocnemius) during faster or uphill gait (Franz et al., 2012; Neptune et al., 2008) is necessary for propulsion but must be appropriately modulated to manage momentum as the trailing leg unloads. Our findings

regarding MD and MDt may represent the kinematic correlates of such adjustments, though direct confirmation would require electromyography (EMG) or kinetic data.

Compared to level gait, where efficient inverted pendulum-like mechanics contribute to minimizing energy cost and supporting step-to-step transitions through effective energy exchange (Donelan et al., 2002; Kuo et al., 2005), the increased the most deceleration of CoM observed during uphill-fast walking in our study, particularly the component directed backward and toward the trailing leg, may reflect a deviation from this economical mechanism. This pattern could represent a less economical but biomechanically necessary movement adjustment aimed at braking momentum under increased mechanical demands, potentially involving increased negative work.

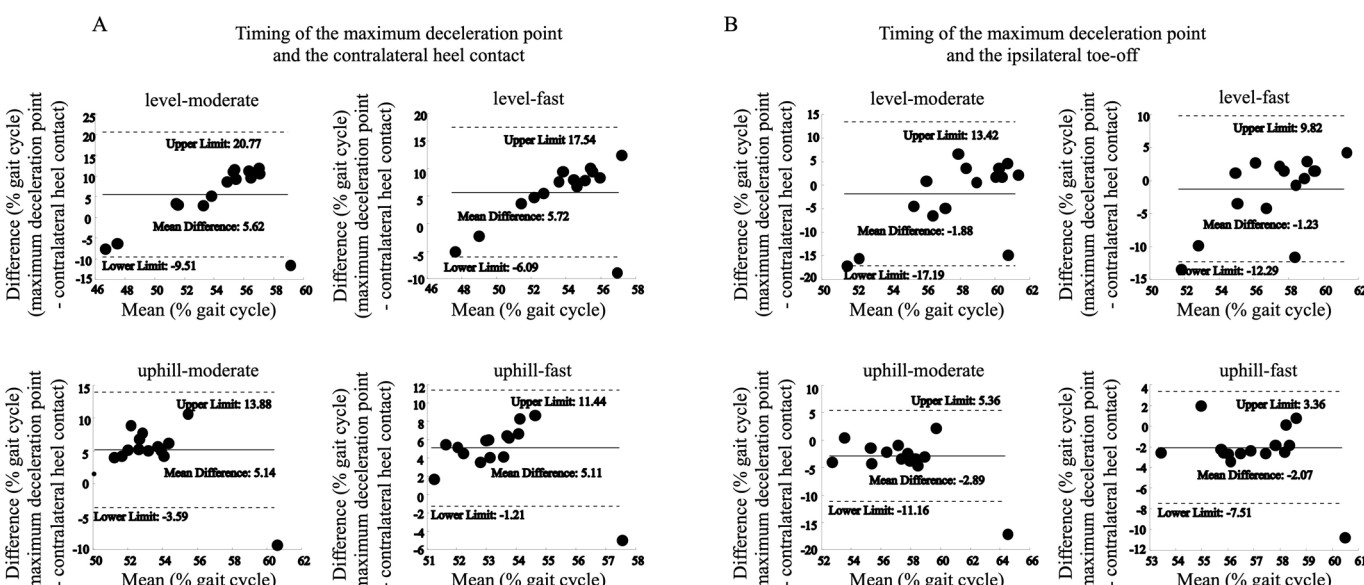

**Fig. 4. Bland–Altman plots comparing the timing of the maximum deceleration point with the contralateral heel contact and ipsilateral toe-off points.** The difference between the maximum deceleration point and contralateral heel contact point across all conditions ranged from 3.22% to 7.72% (mean difference in Bland–Altman analysis: 5.11–7.20; A). In contrast, the difference between the maximum deceleration point and ipsilateral toe-off point ranged from 1.23% to 2.89% across all conditions (mean difference in Bland-Altman analysis: −2.89 to −1.23), indicating that the maximum deceleration point occurred much closer to the ipsilateral toe-off point than to the contralateral heel contact point (B).

In summary, this study demonstrates that healthy young adults systematically modulate the most deceleration and its timing of peak CoM in the horizontal plane during double support when adapting to increased walking speed and incline. These findings contribute to the fundamental understanding of locomotor adaptation mechanisms. Although direct clinical translation requires caution due to the specific population studied (healthy young adults) and the experimental methodology (treadmill walking without kinetic or EMG data), understanding these adaptive kinematic patterns provides an essential baseline. Future investigations should explore whether similar or different patterns of CoM deceleration adjustment are employed by clinical populations (e.g. older adults at risk of falls, individuals post-stroke) and how these patterns may relate to their capacity for gait modulation or response to mechanical challenges. Addressing the limitations of this study, such as incorporating kinetic and EMG measures, examining downhill walking, testing a broader range of speeds and inclines, and including diverse participant groups, will be crucial for developing a more comprehensive understanding and potentially informing clinical applications related to gait assessment, rehabilitation, or fall prevention.

### Limitations

This study has several limitations that warrant consideration when interpreting the findings.

First, we did not measure ground reaction forces. The absence of kinetic data limits our ability to perform a comprehensive analysis of the underlying forces and moments driving the observed CoM movements. Consequently, our interpretations regarding mechanical work, force production, and the specific contributions of braking and propulsive forces remain speculative and based primarily on kinematic observations. Future studies incorporating force plates or instrumented treadmills are needed to elucidate the kinetic mechanisms underlying the observed adaptations in CoM deceleration.

Second, our investigation focused on a limited range of gait speeds (moderate and fast) and a single uphill incline (+6°). Specifically, the walking speeds of 0.83 m/s and 1.0 m/s were chosen to ensure consistency across participants. However, these speeds fall below the typical preferred walking speeds observed in healthy young adult males, and both lie within a relatively narrow functional range. As such, the limited variation in gait speed may constrain the generalizability of our findings and reduce the ability to fully interpret the effects of speed on CoM dynamics. This narrow scope may also not capture the full spectrum of gait adaptations that occur at slower speeds, steeper inclines, or during downhill walking. Future research should explore a broader range of walking speeds and inclines to better reflect natural walking conditions and provide a more comprehensive understanding of how these variables interactively influence CoM dynamics during double support.

Third, the participant sample consisted solely of healthy young adults. While this provides valuable baseline data, the homogeneity limits the direct applicability of our findings to other populations, such as older adults who often exhibit altered gait patterns and increased fall risk, or individuals with specific gait impairments (e.g. neurological or musculoskeletal conditions). Studies involving diverse age groups and clinical populations are necessary to determine whether similar patterns of CoM deceleration modulation are utilized and how they might relate to pathology or rehabilitation outcomes.

Fourth, we did not assess muscle activity using EMG. This limits our ability to directly infer the specific neuromuscular strategies employed during the different gait conditions. While our

kinematic findings (particularly the modulation of MD and MDt) may suggest associated neuromuscular adaptations, incorporating EMG measurements in future studies would provide crucial insights into the relationship between muscle activation patterns and the observed changes in CoM dynamics.

Fifth, the experiment was conducted on a treadmill without explicit visual feedback manipulations or external balance perturbations. Treadmill walking can differ from overground walking, and the predictable laboratory environment may not fully represent the challenges encountered in daily life. Assessing gait adaptations in more ecologically valid, real-world environments or introducing controlled perturbations could provide a more comprehensive understanding of how individuals respond to external challenges during locomotion.

Finally, it is important to reiterate that this study relied solely on kinematic data. While CoM kinematics provide valuable information about movement outcomes, directly inferring underlying control mechanisms or quantifying dynamic stability based solely on these measures has inherent limitations. Conclusions regarding 'stability' or 'control' should be interpreted with caution, as they are inferred from observable movement patterns and not from direct physiological or kinetic evidence.

Addressing these limitations in future research will be essential for building a more robust and physiologically grounded understanding of human gait adaptation across diverse conditions and populations.

### Conclusion

This study demonstrated that healthy young adults systematically modulate the deceleration patterns of their CoM in the horizontal plane during the double-support phase in response to changes in gait speed and incline. Specifically, we found that both faster walking speeds and uphill conditions led to an increase the most deceleration and shifted its timing to occur earlier within the gait cycle.

These findings highlight that adjustments in CoM deceleration near the time of toe-off are closely associated with how gait adapts to biomechanical demands such as increased speed or incline. The observed kinematic adjustments, particularly the earlier timing of deceleration, may reflect movement adaptations that contribute to maintaining coordination and momentum regulation across steps. However, further research incorporating kinetic and EMG data is needed to confirm the underlying neuromuscular mechanisms.

The interplay between speed and incline influences these horizontal plane dynamics, suggesting that multiple biomechanical factors interact to shape gait responses. While derived from healthy young adults, these insights into fundamental gait adaptation mechanisms may help guide future research exploring how individuals adjust gait in response to mechanical demands and could potentially contribute to developing targeted interventions for individuals with gait impairments.

### MATERIALS AND METHODS
#### Participants and experiment

Fifteen healthy young adults (nine females, six males; age 26.1±5.2 years) participated in the current study. Parts of the kinematic data were sourced from our previous study (Kanda et al., 2025). Participants were included if they reported no history of major lower extremity or trunk musculoskeletal injuries or surgeries within the past year, no known neurological, cardiovascular, or respiratory diseases, and no balance or gait impairments that could affect their performance in the experiment. Participants walked on a treadmill (IP-HPM02, H/P/Cosmos Sports & Medical GmbH, Germany) under four combinations of speed and incline, selected to represent common walking conditions. The speeds chosen were

Biology Open

moderate (0.83 m/s, representing a typical comfortable walking speed for young adults) and fast (1.0 m/s, representing a brisker walking pace). The inclines were level (0°) and a moderate uphill slope (+6°). The +6° incline was selected to reflect typical gradients encountered in everyday walking environments, such as sloped sidewalks or ramps. This angle has been commonly used in prior biomechanical studies to evoke meaningful gait adaptations while minimizing excessive physiological strain (Franz et al., 2012). Importantly, pilot testing confirmed that this incline level does not lead to marked fatigue in either male or female participants over the duration of the task, thereby helping to minimize sex-related differences in exertional response. Participants walked for approximately 3 min in each of the four conditions (level-moderate, level-fast, uphill-moderate, uphill-fast) in random order. A motion capture system equipped with nine infrared cameras (100 Hz, Vicon Motion Systems, UK) was used to acquire three-dimensional marker data (Fig. 1A). Reflective markers were placed on anatomical landmarks according to the Plug-in Gait Full Body AI model protocol (Fig. 1B). This model was then used within the Vicon software to calculate the three-dimensional coordinates of body segments and the CoM.

## Data analysis

For the uphill conditions, the raw coordinate data from the laboratory frame (X′: mediolateral, Y′: anterior–posterior along the lab floor, Z′: vertical) were rotated to align with the treadmill surface. Specifically, the anteroposterior (AP) and vertical (VT) coordinates were transformed using standard rotation equations based on the known incline angle ($\theta=6°$) to yield coordinates parallel (AP_slope) and perpendicular (VT_slope) to the walking surface (e.g. $AP\_slope = Y'\cos\theta + Z'\sin\theta$; $VT\_slope = -Y'\sin\theta + Z'\cos\theta$). The mediolateral (ML) coordinate (X′) remained unchanged. This ensured that the AP direction was consistently defined along the path of progression for both level and uphill conditions (Fig. 1C). Subsequent analyses involving the horizontal plane used the ML and AP_slope coordinates. Marker coordinates were processed with a second-order Butterworth low-pass filter at 10 Hz, based on previous validation studies that demonstrated this filter's effectiveness in minimizing high-frequency noise while preserving the biomechanical signal of interest during dynamic tasks (Koltermann et al., 2018). This combination was shown to provide an optimal balance between signal fidelity and noise reduction, particularly in human movement analysis. Following previous research (Roerdink et al., 2008), the gait cycle was defined by heel contact and toe-off from the position and velocity of the heel marker, and data from 100 right steps were analyzed.

The variables used for analysis were the double support time ratio (DST), step length (SL), the most deceleration of CoM in the horizontal plane (MD), and its timing (MDt). MD was calculated as follows. The instantaneous CoM velocity vector in the horizontal plane (combining anteroposterior and mediolateral components) was calculated, and horizontal-plane acceleration was obtained by numerical differentiation. The most negative value of this acceleration during the double-support phase was extracted as MD, representing the largest deceleration acting opposite to the direction of movement. MDt was defined as the time point at which MD occurred, normalized to 100% of the corresponding gait cycle duration (from right heel contact to the next right heel contact).

## Statistical methods

Normality of all variables was tested using the Lilliefors test. Outliers (values exceeding two standard deviations from the mean) were excluded. To explore potential sex-related effects, independent-samples $t$-tests were conducted to compare female and male participants for each of the primary outcome variables (MD, MDt, SL, and DST) under all four gait conditions. While not sufficiently powered for formal inference, this exploratory analysis was included for transparency. To compare the conditions, a repeated measures ANOVA was performed, accounting for the within-subject design. When the assumption of sphericity was violated, Greenhouse-Geisser correction was applied.

For post-hoc comparisons, Tukey's multiple comparison test was used when significant differences were found. Additionally, the agreement between the MDt and the ipsilateral heel contact point (iHC) or contralateral

toe-off point (cTO) was assessed using the Bland–Altman method, with bias, limits of agreement (LoA), and their confidence intervals reported. All statistical analyses were conducted using MATLAB (2024a, MathWorks Inc.), with a significance level set at 5%.

## Competing interests

The authors declare no competing or financial interests.

## Author contributions

Conceptualization: K.H.; Data curation: S.T., N.K., K.H.; Formal analysis: S.T., N.K., K.H.; Funding acquisition: K.H.; Investigation: S.T., N.K., K.H.; Methodology: S.T., K.H.; Project administration: S.T., K.H.; Resources: S.T., K.H.; Software: S.T., N.K., K.H.; Supervision: S.T., N.K., K.H.; Validation: S.T., N.K., K.H.; Visualization: S.T., N.K., K.H.; Writing – original draft: S.T., K.H.; Writing – review & editing: S.T., N.K., K.H.

## Funding

This work was supported by a Grant-in-Aid for JSPS Research, Fellows 23K16625. Open Access funding provided by Japan Society for the Promotion of Science. Deposited in PMC for immediate release.

## Data and resource availability

The datasets supporting this article have been uploaded as part of the Supplementary Material. Source data for this study are not publicly available due to privacy or ethical restrictions. The source data are available to verified researchers upon request by contacting the corresponding author.

## Ethical statement

The experiment was planned in accordance with the Declaration of Helsinki and was conducted after obtaining approval from the Ethics Committee of the affiliated institution (number 29501). Written informed consent was obtained from all participants.

## Peer review history

The peer review history is available online at https://journals.biologists.com/bio/article-pdf/14/7/bio062037/3655497/bio062037_review_history.pdf.

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
