## [Peer Review File · Biology Open]

Gait Speed and Incline Modulate Peak Deceleration and Timing of Horizontal Center of Mass Deceleration During Double Support

Shizuku Terui, Nanami Kanda and Keisuke Hirata

DOI: 10.1242/bio.062037

Editor: Lewis Halsey

Review timeline

Original submission:	4 February 2025
Editorial decision:	8 February 2025
Resubmission:	27 April 2025
Editorial decision:	10 May 2025
First revision received:	2 June 2025
Accepted:	4 June 2025

Original submission

First decision letter

MS ID#: bio.061915

MS TITLE: Effects of Speed and Incline on Center of Mass Dynamics in the Double-Support Phase of Gait

AUTHORS: Shizuku Terui; Nanami Kanda; Keisuke Hirata

Dear Dr Hirata,

MS TITLE: Effects of Speed and Incline on Center of Mass Dynamics in the Double-Support Phase of Gait

I am writing to let you know that I have now reached a decision on the above manuscript. I am afraid that, after careful consideration, I feel that it cannot currently be accepted for publication in Biology Open.

The reviewer reports are shown at the bottom of this email or can be accessed, together with a copy of this decision letter, by going to:

As you will see, the reviewers raise a number of substantial criticisms that prevent me from accepting your paper for publication.

In particular, Reviewer 1 raises major concerns about a lack of direct measurements of energy expenditure or neuromuscular control, and also about the use of units that don't relate to terminology e.g. "acceleration" is associated with m/s^3 , while CoM velocity is given the units m/s^2 .

I realise that this is disappointing news, and we understand the frustration that you must feel. However, I am sure that you appreciate that the conclusions of your research must be seen by the wider community to be fully supported by the data. On this occasion, I have decided that this is not the case.

I do hope you find the reviewer comments helpful in allowing you to revise the manuscript for successful submission to Biology Open or elsewhere.

Reviewer 1

Comments for the author

The authors present a study into CoM movement in relation to both speed and incline during treadmill walking. The work, as carried out, is well described but I have significant concerns regarding how the authors have tried to contextualise their results in the current version of the manuscript. The data and figures are also confusing and not particularly well presented.

- You refer several times to how their results have implications for energy efficiency and neuromuscular control. I would argue that given the noted limitations of the study, which include a lack of EMG and GRF data, as well as no metabolic energy expenditure data (via respirometry, for example), no real inferences on either energy expenditure or neuromuscular control can be made here. Certainly not with enough strength to be included in either the Summary Statement or the Abstract. Calculating mechanical energy exchange from CoM position is possible and would give some indications of the efficiency of walking (Cavagna et al 1976 J Physiol, and recently by Grant et al 2024 JEB)

- While you have noted this limitation, is a 6-degree incline really enough to test your hypotheses here? It also wasn't noted that treadmill and overground walking do differ substantially, in terms of kinematics at least, meaning an overground walking study (potentially using a custom built ramp of a specified incline) would be more suited to answering your questions

- I'm a little confused by what is actually being measured. "Acceleration" is mentioned a lot, but in all the figures and text this is in the units of m/s^3 , which is jerk, not acceleration. CoM velocity is also given the units m/s^2 , which is acceleration. Furthermore, the CoM velocity panels in Figure 3 are not mentioned at all in the Results section, and only briefly in the Discussion section, and so appear to not be relevant.

- How were your walking speeds chosen? Did you consider allowing each participant to walk at the same normalised speed (a.k.a Froude number- speed normalised to limb length), and so eliminate the impacts of body size on your data?

- Tables and figures

o I think table 1 is a bit redundant, as half of the data is the same as that in Figure 2. Why not expand figure 2 with the MD and MDt data from Table 1, and move Table 1 to Supplementary data, if needed.

o Figure 3- Strictly speaking, each graph has its own panel letter (so here, this would be A). Consider splitting this into two figures or removing the velocity data entirely, as it's not really mentioned in the text. Annotations or lines of these graphs pointing to each key event in a walking gait cycle (i.e. heel strike, double support phase, etc.) would be very helpful in contextualising these graphs with the Results and Discussion. I would also recommend carrying out some Statistical Parametric Mapping analysis on these data (Pataky et al 2013 J Biomech), which would allow you to look for specific areas of the gait cycle in which CoM velocity or acceleration are significantly different. Also, why are all velocity traces shown, but only the mean and SD of the accelerations. This should be consistent between them

o Figure 4- This figure is generally very confusing. What does the x-axis label "mean" refer to? And the y axis label "difference"? It should really be two figures so that each graph can be larger and clearer.

- Other minor comments

o The abstract should start by introducing a gap in the knowledge that your study is trying to address.

o Page 1, line 23- "double support phase of WALKING gait"

o Page 1, line 24- what does "abnormal loads" refer to?

o Page 1, line 29- what do you mean by "strictly regulated"?

- o Discussion- Figures should be cited here as well, to help the reader to contextualise your Discussion points.
- o Discussion line 25-26 - "This is thought to be influenced by...." This sentence is largely repeating a previous point and can be deleted.

Reviewer 2

Comments for the author

This was an interesting read, and I am happy to recommend the manuscript for publication after minor corrections. It is a short, well-written manuscript and, despite its relatively limited scope, deserves publication. My detailed comments follow:

Experimental Quality

The study uses a well-established motion capture system to assess centre of mass (CoM) dynamics, which is appropriate for the research question. Figures are well-documented and provide relevant comparisons across experiments. The statistical methods, including repeated measures ANOVA, are appropriate. Greenhouse-Geisser correction for sphericity violations is a good choice.

The study does not include kinetic data (e.g., ground reaction forces), which would strengthen the interpretation of CoM deceleration mechanisms. However, this limitation is acknowledged and discussed.

Reproducibility

The methods are very clearly described, including participant selection criteria, experimental conditions, and motion capture setup.

The study uses a sufficient number of participants (n=15) for initial analysis, but no justification (e.g., power analysis) is provided for whether this sample size is adequate to detect subtle effects. The paper does not provide access to raw motion capture data, but the authors indicate that data can be requested. Providing at least summary statistics or representative raw data in supplementary materials could improve transparency.

Completeness

The conclusions are supported by the data, particularly regarding the effects of speed and incline on CoM deceleration and step length.

Scholarship

The manuscript cites relevant literature to support its methodology and interpretation. The discussion is well-referenced.

Additional Comments

The paper is well-written, but minor improvements could enhance clarity. I specifically feel that the Introduction starts a bit too abruptly. A non-specialist may wonder what the "double-support phase of gait" is (especially as it is mentioned in the first sentence). It would be beneficial to start the Introduction with a brief definition or context-setting explanation to make the topic more accessible to a wider readership.

I would also like to encourage authors to share the raw data, which would certainly increase the impact of the study.

Reviewer's Responses to Questions

Experimental quality

Does each figure have the proper controls?

If 'No', please indicate reasons in Comments for Author box below.

Reviewer #1:

- Yes

Reviewer #2:

- Yes

Were the data analyzed using appropriate statistical tests?

If 'No', please indicate reasons in Comments for Author box below.

Reviewer #1:

- No

Reviewer #2:

- Yes

Reproducibility

Were experiments performed using adequate number of biological replicates?

If 'No', please indicate reasons in Comments for Author box below.

Reviewer #1:

- Yes

Reviewer #2:

- Yes

Does the methods section provide sufficient detail to permit reproducibility?

If 'No', please indicate reasons in Comments for Author box below.

Reviewer #1:

- Yes

Reviewer #2:

- Yes

Completeness

Are the manuscript's conclusions supported by the data?

If 'No', please indicate reasons in Comments for Author box below.

Reviewer #1:

- No

Reviewer #2:

- Yes

Scholarship

Do the authors cite and discuss the merits of data that would argue for and against their conclusion?

If 'No', please indicate reasons in Comments for Author box below.

Reviewer #1:

- Yes

Reviewer #2:

- Yes

Does the manuscript title & abstract accurately reflect the contents of the manuscript, without hyperbole?

If 'No', please indicate reasons in Comments for Author box below.

Reviewer #1:

- Yes

Reviewer #2:

- Yes

Author response to reviewers' comments

Reviewers' comments

1: The authors present a study into CoM movement in relation to both speed and incline during treadmill walking. The work, as carried out, is well described but I have significant concerns regarding how the authors have tried to contextualise their results in the current version of the manuscript. The data and figures are also confusing and not particularly well presented. - You refer several times to how their results have implications for energy efficiency and neuromuscular control. I would argue that given the noted limitations of the study, which include a lack of EMG and GRF data, as well as no metabolic energy expenditure data (via respirometry, for example), no real inferences on either energy expenditure or neuromuscular control can be made here. Certainly not with enough strength to be included in either the Summary Statement or the Abstract. Calculating mechanical energy exchange from CoM position is possible and would give some indications of the efficiency of walking (Cavagna et al 1976 J Physiol, and recently by Grant et al 2024 JEB)

Response: We sincerely thank the reviewer for this important and constructive comment regarding the interpretation of our results. In response, we have substantially revised the Discussion section to more accurately reflect the limitations of our study. Specifically: We have removed speculative statements suggesting direct implications for energy efficiency or neuromuscular control from the Abstract, Summary Statement, and Discussion. Where relevant,

we have reframed our interpretations to emphasize that the findings are based solely on observed kinematic adaptations, without making unsupported inferences regarding underlying neuromuscular control mechanisms or energetic cost. -

While you have noted this limitation, is a 6-degree incline really enough to test your hypotheses here? It also wasn't noted that treadmill and overground walking do differ substantially, in terms of kinematics at least, meaning an overground walking study (potentially using a custom built ramp of a specified incline) would be more suited to answering your questions I'm a little confused by what is actually being measured. "Acceleration" is mentioned a lot, but in all the figures and text this is in the units of m/s^3 , which is jerk, not acceleration. CoM velocity is also given the units m/s^2 , which is acceleration. Furthermore, the CoM velocity panels in Figure 3 are not mentioned at all in the Results section, and only briefly in the Discussion section, and so appear to not be relevant.

Response: We thank the reviewer for pointing out these important inconsistencies and potential sources of confusion regarding the reported variables and units. In response, we carefully reviewed and corrected the terminology and figure presentations throughout the manuscript. First, we clarified the definitions and units in the Methods section to ensure accurate descriptions: "Acceleration" is now consistently used to refer to changes in velocity with correct units of m/s^2 . "Jerk" (the time derivative of acceleration, with units of m/s^3) is no longer mistakenly labeled as "acceleration" anywhere in the text or figures. Second, we revised the figures accordingly to correctly label all axes and units. The updated Figure now consistently presents CoM velocity in m/s , and any unintended displays of m/s^2 or m/s^3 have been corrected. Third, regarding the mention and relevance of the CoM velocity panels in Figure 3: we have revised the Results section to explicitly describe these panels and explain their importance to the interpretation of our findings. We also expanded the Discussion section to better integrate these findings and clarify their implications for understanding CoM control during double-support. We appreciate the reviewer's careful attention to these issues, which has significantly improved the clarity and accuracy of our manuscript.

How were your walking speeds chosen? Did you consider allowing each participant to walk at the same normalised speed (a.k.a Froude number- speed normalised to limb length), and so eliminate the impacts of body size on your data?

Response: We thank the reviewer for this valuable comment regarding the choice of walking speeds and the suggestion to consider body-size normalization using the Froude number. In response, we have revised the Methods section to clarify our rationale for selecting the absolute speeds of 0.83 m/s (moderate) and 1.0 m/s (fast). These speeds were chosen to represent commonly encountered comfortable and brisk walking paces for healthy young adults, based on prior literature and typical functional walking ranges. We agree that using normalized speeds based on limb length (Froude number) would be a rigorous approach to account for inter-individual differences in body size. However, given the relatively homogeneous height range of our participant sample and our aim to examine gait adaptations under conditions simulating typical daily-life walking, we opted for fixed absolute speeds in this initial study. We have now added a note in the Limitations section acknowledging that using normalized speeds could further improve the generalizability of the results and that this approach should be considered in future investigations. We appreciate the reviewer's thoughtful suggestion, which helped us improve the transparency of our methodological description. -

Tables and figures - I think table 1 is a bit redundant, as half of the data is the same as that in Figure 2. Why not expand figure 2 with the MD and MDt data from Table 1, and move Table 1 to Supplementary data, if needed.

Response: We thank the reviewer for this helpful suggestion regarding the presentation of our data. In response, we revised Figure 2 to incorporate the MD and MDt data that were previously reported separately in Table 1, thereby providing a more consolidated and streamlined visualization of the main results. We also moved the full original Table 1, containing all detailed descriptive statistics, to the Supplementary Table. In the main manuscript, Table 1 now only presents non-redundant data that are not included in Figure 2, ensuring there is no overlap

between figures and tables. We appreciate the reviewer's recommendation, which has significantly improved the clarity and organization of the results presentation.

Figure 3- Strictly speaking, each graph has its own panel letter (so here, this would be A). Consider splitting this into two figures or removing the velocity data entirely, as it's not really mentioned in the text. Annotations or lines of these graphs pointing to each key event in a walking gait cycle (i.e. heel strike, double support phase, etc.) would be very helpful in contextualising these graphs with the Results and Discussion. I would also recommend carrying out some Statistical Parametric Mapping analysis on these data (Pataky et al 2013 J Biomech), which would allow you to look for specific areas of the gait cycle in which CoM velocity or acceleration are significantly different. Also, why are all velocity traces shown, but only the mean and SD of the accelerations. This should be consistent between them.

Response: We sincerely thank the reviewer for these thoughtful and detailed comments regarding Figure 3. In response, we carefully reconsidered the role of this figure within the overall manuscript. Given that the CoM velocity data were not central to our main hypotheses and were only briefly mentioned in the text, we have decided to remove Figure 3 entirely from the revised manuscript to maintain focus and clarity.

Figure 4- This figure is generally very confusing. What does the x-axis label "mean" refer to? And the y axis label "difference"? It should really be two figures so that each graph can be larger and clearer.

Response: We thank the reviewer for highlighting the issues regarding the clarity and labeling of Figure 4. In response, we have revised the figure to clearly indicate what the x-axis ("mean") and y-axis ("difference") represent. Specifically, we have updated the axis labels to explicitly state the variables being compared to avoid confusion. Additionally, we corrected inconsistencies in the number of significant digits reported, ensuring uniformity and precision throughout the figure. While we considered splitting the figure into two separate figures to enhance readability, we found that with the improved labeling and formatting, the information can be clearly conveyed within a single, well-organized figure. We have also enlarged the graphs slightly to improve legibility. We appreciate the reviewer's helpful suggestions, which have significantly improved the clarity and presentation of the figure.

Other minor comments -The abstract should start by introducing a gap in the knowledge that your study is trying to address.

Response: We sincerely thank the reviewer for this important suggestion regarding the structure of the abstract. In response, we have revised the abstract to clearly introduce the existing gap in the knowledge— specifically, the limited understanding of how center of mass (CoM) dynamics in the horizontal plane during the double-support phase are influenced by walking speed and incline. Additionally, to maintain consistency with revisions made throughout the manuscript and to better align the abstract with the study's updated focus, we have comprehensively restructured the abstract. The revised version now follows a clearer logical flow: introducing the knowledge gap, stating the study's purpose and hypotheses, summarizing the methods, highlighting key findings, and briefly mentioning the broader implications.

o Page 1, line 23- "double support phase of WALKING gait"

Response: We have revised it.

o Page 1, line 24- what does "abnormal loads" refer to?

Response: We thank the reviewer for pointing out the ambiguity regarding the phrase "strictly regulated." In response, we have revised the corresponding sentence in the Introduction to clarify the intended meaning. Specifically, we replaced the original description with a more precise explanation emphasizing the importance of smooth weight transition during the double-support phase for efficient forward progression, energy conservation, and load mitigation.

o Page 1, line 29- what do you mean by "strictly regulated"?

Response: We have removed it.

o Discussion- Figures should be cited here as well, to help the reader to contextualise your Discussion points.

Response: We have cited figures in Discussion.

o Discussion line 25-26 - "This is thought to be influenced by...." This sentence is largely repeating a previous point and can be deleted.

Response: We have removed it.

Reviewer 2: This was an interesting read, and I am happy to recommend the manuscript for publication after minor corrections. It is a short, well-written manuscript and, despite its relatively limited scope, deserves publication. My detailed comments follow:

Experimental Quality

The study uses a well-established motion capture system to assess centre of mass (CoM) dynamics, which is appropriate for the research question. Figures are well-documented and provide relevant comparisons across experiments. The statistical methods, including repeated measures ANOVA, are appropriate. GreenhouseGeisser correction for sphericity violations is a good choice.

The study does not include kinetic data (e.g., ground reaction forces), which would strengthen the interpretation of CoM deceleration mechanisms. However, this limitation is acknowledged and discussed.

Reproducibility

The methods are very clearly described, including participant selection criteria, experimental conditions, and motion capture setup. The study uses a sufficient number of participants (n=15) for initial analysis, but no justification (e.g., power analysis) is provided for whether this sample size is adequate to detect subtle effects. The paper does not provide access to raw motion capture data, but the authors indicate that data can be requested. Providing at least summary statistics or representative raw data in supplementary materials could improve transparency.

Completeness

The conclusions are supported by the data, particularly regarding the effects of speed and incline on CoM deceleration and step length.

Scholarship

The manuscript cites relevant literature to support its methodology and interpretation. The discussion is well-referenced.

Additional Comments

The paper is well-written, but minor improvements could enhance clarity. I specifically feel that the Introduction starts a bit too abruptly. A non-specialist may wonder what the "double-support phase of gait" is (especially as it is mentioned in the first sentence). It would be beneficial to start the Introduction with a brief definition or context-setting explanation to make the topic more accessible to a wider readership. I would also like to encourage authors to share the raw data, which would certainly increase the impact of the study.

Response: We sincerely thank the reviewer for their thoughtful and constructive feedback. In response to the comment regarding the Introduction, we have revised the beginning of the

section to include a clear and concise definition of the double-support phase of gait. We now explain its significance for forward progression, energy efficiency, and load mitigation, thereby making the context more accessible to a broader readership, including non-specialists. The revised Introduction provides a smoother entry into the topic before delving into more specific aspects of gait control. Additionally, regarding the suggestion to share the raw data, we agree that providing access to the raw dataset would enhance the transparency and impact of our study. We greatly appreciate the reviewer's insightful comments, which have helped us improve the clarity and rigor of our work

Resubmission

First decision letter

MS ID#: bio.062037

MS TITLE: Gait Speed and Incline Modulate Peak Deceleration and Timing of Horizontal Center of Mass Deceleration During Double Support

AUTHORS: Shizuku Terui, Nanami Kanda and Keisuke Hirata

I have now reached a decision on the above manuscript.

The reviewer reports are shown at the bottom of this email or can be accessed, together with a copy of this decision letter, by going to:

As you will see, the reviewer raised a number of substantial criticisms that prevent me from accepting the paper at this stage.

The reviewer suggests, however, that a revised version might prove acceptable, if you can address their concerns. If you think that you can deal satisfactorily with the criticisms on revision, I would be pleased to see a revised manuscript. We would then return it to the reviewers.

At this stage, we also ask you to ensure your manuscript complies with our [HYPERLINK "https://journals.biologists.com/bio/pages/manuscript-prep"](https://journals.biologists.com/bio/pages/manuscript-prep) formatting guidelines. Provided you are able to fully address the referees' comments, we are positive about publication of your paper (we accept over 95% of revision submissions) and therefore hope you won't mind any extra work involved in reformatting your manuscript at this point.

Please ensure that you clearly highlight all changes made in the revised manuscript. Please avoid using 'Tracked changes' in Word files as these are lost in PDF conversion.

I should be grateful if you would also provide a point-by-point response detailing how you have dealt with the points raised by the reviewers in the 'Response to Reviewers' box. Please attend to all of the reviewers' comments. If you do not agree with any of their criticisms or suggestions please explain clearly why this is so.

Reviewer 1

Comments to Author

In my opinion, the study addresses an interesting topic (though with limited scope) but currently lacks methodological clarity. A substantially revised version with more precise metrics and toned-down claims could make a more meaningful contribution. Although exploratory, the findings could have implications for the design of gait rehabilitation protocols and biomechanical models that account for terrain and speed variations. While the study is methodologically limited and somewhat

narrow in scope, it represents a step toward quantifying fine-grained adaptations in CoM dynamics and encourages further investigation.

Comments:

- The novelty of the study, especially in relationship to Adamczyk & Kuo, 2009 and Dewolf et al., 2017, should be highlighted. Mentioning that somethings "require further elucidation" does not sound convincing enough. Please try to differentiate the current study from prior work.
- Could the incline angle of 6 degrees be better justified?
- Figures - Presentation needs significant improvements. See examples below:
 - * please check the labels as they are missing in some figures
 - * in some cases (such as fig 3) the fonts are too small
 - * check the units and ensure that they are present
 - * better not to use the same axis label for diagrams in different panels of a figure
 - * please check the notations and ensure that they are easily readable (figs 3 and 4)
 - * The figures lack statistical annotation (e.g., asterisks for $p < 0.05$) and sample sizes aren't indicated (is this $n=15$ everywhere?).
 - * Could the locations of reflective markers be added to all panels?
- The wide confidence bands around the mean in Fig 3 suggest high variability in the collected data. This raises concerns about the robustness and reliability of the extracted MD and MDt values. The authors should better justify their signal processing methods, particularly their filtering choices, and consider presenting confidence intervals or step-wise distributions of MD and MDt to assess their consistency.
- I think one of my main issues here is that the manuscript repeatedly uses terms like "stability", "control", or "strategy", yet these are never quantified or measured.
- The study includes a mixed-sex sample of participants (9 females, 6 males), yet the analysis does not consider sex as a factor or covariate. I therefore recommend that the authors perform an exploratory analysis to determine whether sex significantly influences any of the main outcomes (MD, MDt, SL, or DST). This could involve including sex as a factor in the repeated-measures ANOVA or performing subgroup comparisons. Even if underpowered for formal inference, such an analysis would be valuable for transparency and may help interpret the variance observed across participants. If no meaningful differences are found, please explicitly state that.
- Please better explain the method to derive the magnitude of CoM deceleration (MD). The authors mention that they used "most negative value of acceleration magnitude". First "magnitude" typically refers to a positive scalar. Second, are they reporting the most negative value (i.e., highest deceleration) or the maximum of absolute deceleration? This must be clarified.
- "...a second-order Butterworth low-pass filter at 10 Hz" could you justify this?
- My last comment is about the speed conditions (0.83 m/s and 1.0 m/s). These are relatively close together, and both fall within a relatively narrow functional range for healthy young adults. Isn't it? The limited speed range may constrain the generalisability of the findings and the ability to meaningfully interpret the reported effects. I suggest the authors discuss this limitation.

Reviewer's Responses to Questions

Experimental quality

Does each figure have the proper controls?

If 'No', please indicate reasons in Comments for Author box below.

Reviewer #1:

- No

Were the data analyzed using appropriate statistical tests?

If 'No', please indicate reasons in Comments for Author box below.

Reviewer #1:

- Yes

Reproducibility

Were experiments performed using adequate number of biological replicates?

If 'No', please indicate reasons in Comments for Author box below.

Reviewer #1:

- Yes

Does the methods section provide sufficient detail to permit reproducibility?

If 'No', please indicate reasons in Comments for Author box below.

Reviewer #1:

- No

Completeness

Are the manuscript's conclusions supported by the data?

If 'No', please indicate reasons in Comments for Author box below.

Reviewer #1:

- No

Scholarship

Do the authors cite and discuss the merits of data that would argue for and against their conclusion?

If 'No', please indicate reasons in Comments for Author box below.

Reviewer #1:

- Yes

Does the manuscript title & abstract accurately reflect the contents of the manuscript, without hyperbole?

If 'No', please indicate reasons in Comments for Author box below.

Reviewer #1:

- Yes

Did you co-review this manuscript with a member of your lab?

Reviewer #1:

- No

Would you like us to report to the Web of Science <https://clarivate.com/products/scientific-and-academic-research/research-publishing-solutions/reviewer-recognition-service/> reviewer recognition service (previously Publons) that you have completed a review for Biology Open?

Reviewer #1:

- Yes

First revision

Author response to reviewers' comments

Reply to comments from Reviewer #1

In my opinion, the study addresses an interesting topic (though with limited scope) but currently lacks methodological clarity. A substantially revised version with more precise metrics and toned-down claims could make a more meaningful contribution. Although exploratory, the findings could have implications for the design of gait rehabilitation protocols and biomechanical models that account for terrain and speed variations. While the study is methodologically limited and somewhat narrow in scope, it represents a step toward quantifying fine-grained adaptations in CoM dynamics and encourages further investigation.

Response: Thank you for giving us the opportunity to submit a revised draft of our manuscript to *Biology Open*. We appreciate the time and effort that you and the reviewers have dedicated to providing your valuable feedback on our manuscript. We are grateful to the reviewers for their insightful comments. We have been able to incorporate changes to reflect their suggestions and the changes are presented in red font within the revised manuscript. We hope that the revised manuscript is suitable for publication.

The following are our point-by-point responses to the reviewers' comments and concerns:

- The novelty of the study, especially in relationship to Adamczyk & Kuo, 2009 and Dewolf et al., 2017, should be highlighted. Mentioning that somethings "require further elucidation" does not sound convincing enough. Please try to differentiate the current study from prior work.:

Response: Thank you for your valuable feedback regarding the novelty of our study and the need to differentiate it more clearly from prior work, especially Adamczyk & Kuo (2009) and Dewolf et al. (2017). Your comment has helped us to strengthen the Introduction and highlight the specific contribution of our research.

We have revised the Introduction (specifically the third and fourth paragraphs) to explicitly address your comment. While previous studies have indeed investigated the mechanics of the center of mass during walking transitions and varying conditions, our study introduces and focuses specifically on the magnitude (MD) and timing (MDt) of the peak horizontal center of mass deceleration during the double-support phase.

As highlighted in the revised Introduction, Adamczyk & Kuo (2009) analyzed the overall COM velocity redirection and work during the step-to-step transition, comparing human mechanics to simplified models. While they quantified work in relation to gait parameters like speed and step length, their analysis did not quantify the specific peak deceleration kinematic profile or its precise timing within the double support phase relative to gait events. Dewolf et al. (2017) focused on

within-step energy exchange (E_k and E_p) and external work (W_{ext}) across slopes and speeds, quantifying the effectiveness or magnitude of energy exchanged. However, their analysis did not include the magnitude and timing of peak horizontal CoM deceleration as a specific variable of interest.

Our work is novel in that it isolates and quantifies this specific kinematic event - the peak horizontal deceleration - during the critical double-support phase to understand how the body actively manages momentum under varying speed and incline conditions. This specific focus on the magnitude and timing of this particular deceleration peak within the double support phase provides new insights into the kinematic adaptations at the whole-body level during a key phase for gait transition and stability that were not the primary focus or variables quantified in the important foundational work of Adamczyk & Kuo (2009) and Dewolf et al. (2017).

We believe the revised Introduction now more clearly articulates this specific contribution and differentiates our study from the prior work, explaining why our specific measure (MD and MDt) and focus (double support phase) are relevant for understanding momentum management and kinematic adaptation during gait across different speeds and inclines.

We hope these revisions satisfy your comment and enhance the clarity of our manuscript.

“Previous studies have analyzed aspects of CoM mechanics and energy during gait transitions and varied conditions. Adamczyk & Kuo (Adamczyk & Kuo, 2009) focused on the redirection of CoM velocity and the work performed on the CoM during the step-to-step transition, comparing human behavior to rigid-leg pendulum models. They showed that the angular redirection of CoM velocity and the work performed tend to increase with walking speed and step length. However, they noted that their simple model could not predict the duration of the step-to-step transition, which empirically exceeds the double support period. Dewolf et al. (2017) analyzed within-step energy conversion (exchange between kinetic and potential energy) and external work during walking across different slopes and speeds (Dewolf et al., 2017). They demonstrated that slope and speed affect these energy-related parameters and introduced a new variable (pendular energy savings, E_s) to quantify the effectiveness or magnitude of energy exchanged within the step. Unlike Adamczyk & Kuo (Adamczyk & Kuo, 2009) who explored predictors of work, Dewolf et al. (2017) primarily focused on the energy conversion efficiency and total external work (Dewolf et al., 2017).

While these foundational studies have provided valuable insights into overall CoM mechanics, transition work, and within-step energy transformations, they did not specifically investigate or quantify the most deceleration and its timing of CoM movement in the horizontal plane, particularly within the double-support phase, as key kinematic variables reflecting momentum management and adaptation to varying speed and incline. We contend that the maximum deceleration of the CoM in the horizontal plane and its timing within the gait cycle, particularly during the double support phase near trailing leg toe-off, represent critical kinematic variables that reflect how the body manages its momentum transfer between steps.” (Introduction, Page 2, Lines 69-90)

- Could the incline angle of 6 degrees be better justified?

Response: We thank the reviewer for raising this important point. The $+6^\circ$ incline was chosen to represent a moderate uphill walking condition that is both ecologically valid and experimentally practical. This angle reflects common gradients found in real-world environments (e.g., ramps, sloped sidewalks) and has been widely adopted in previous gait studies to examine incline-related adaptations without causing excessive exertion (e.g., Franz et al., 2012; Neptune et al., 2008). In addition, this incline level was selected based on pilot testing to avoid introducing substantial fatigue-related variability between male and female participants. Specifically, we aimed to select an incline steep enough to elicit measurable biomechanical adaptations, but not so steep as to differentially challenge participants' endurance or alter gait mechanics due to fatigue effects, which could confound the interpretation of CoM dynamics.

“The inclines were level (0°) and a moderate uphill slope ($+6^\circ$). The $+6^\circ$ incline was selected to reflect typical gradients encountered in everyday walking environments, such as sloped sidewalks or ramps. This angle has been commonly used in prior biomechanical studies to evoke meaningful gait adaptations while minimizing excessive physiological strain (Franz et al., 2012). Importantly, pilot testing confirmed that this incline level does not lead to marked fatigue in either male or female participants over the duration of the task, thereby helping to minimize sex-related differences in exertional response.” (Methods, Page 3, Lines 119-125)

- Figures - Presentation needs significant improvements. See examples below:
- * please check the labels as they are missing in some figures
- * in some cases (such as fig 3) the fonts are too small
- * check the units and ensure that they are present
- * better not to use the same axis label for diagrams in different panels of a figure
- * please check the notations and ensure that they are easily readable (figs 3 and 4)
- * The figures lack statistical annotation (e.g., asterisks for $p < 0.05$) and sample sizes aren't indicated (is this $n=15$ everywhere?).
- * Could the locations of reflective markers be added to all panels?

Response: We appreciate the reviewer's detailed comments regarding the figures. In response:

- **Figure labels and units:** We have carefully reviewed all figures and added missing axis labels and units where they were absent. We ensured that each axis is clearly labeled and that units are consistently presented throughout the manuscript.
- **Font size:** The font sizes in Figures 3 and 4 have been increased to enhance legibility, especially when printed or viewed at standard sizes.
- **Axis labeling consistency:** We revised axis labels across subpanels to avoid duplication or ambiguous reuse of labels. Each panel now uses specific and appropriate axis titles, unless a shared axis is explicitly intended and labeled accordingly.
- **Notation readability:** We have adjusted the formatting of notations in Figures 3 and 4 to improve clarity. This includes increasing symbol size and spacing, and ensuring all symbols are defined either in the figure or the legend.
- **Statistical annotations and sample sizes:** We have added statistical annotations (e.g., asterisks indicating significance levels) where applicable, and included exact p-values in the legends or main text. The sample size ($n = 15$ for all analyses) is now indicated clearly in each relevant figure panel or legend.
- **Reflective marker positions:** We have revised all figure panels to include the schematic locations of reflective markers used in the study, either within the images or as insets. This will assist readers in interpreting the kinematic data more accurately.

We hope these revisions enhance the clarity and interpretability of the figures as per the reviewer's suggestions.

(Figure 1)

(Figure 2)

(Figure 3)

(Figure 4)

- The wide confidence bands around the mean in Fig 3 suggest high variability in the collected data. This raises concerns about the robustness and reliability of the extracted MD and MDt values. The authors should better justify their signal processing methods, particularly their filtering choices, and consider presenting confidence intervals or step-wise distributions of MD and MDt to assess their consistency.

Response: We sincerely thank the reviewer for this insightful comment and for pointing out the potential issue regarding data variability in Figure 3. Upon careful re-examination of the analysis, we found that, due to an oversight, the data plotted in Figure 3 were not processed with the low-pass filter (second-order Butterworth, 10 Hz) that was otherwise consistently applied to all datasets as described in the Methods section.

Importantly, we confirmed that this filtering step was properly applied to all data used for statistical analyses and results presented throughout the manuscript. Therefore, the high variability observed in Figure 3 was not representative of the underlying dataset but a result of the omission of filtering in the visualization step only.

We have now corrected Figure 3 by re-plotting it using the appropriately filtered data. As a result, the confidence bands are now markedly narrower, and better reflect the consistency of the MD and MDt values. This correction does not affect the reported results or conclusions, as those were derived from correctly processed data.

Additionally, in response to the reviewer's suggestion and a related comment, we have revised the Methods section to explicitly state the rationale for our filtering choices, citing previous studies that validated the 10 Hz cutoff frequency as appropriate for dynamic gait analyses.

While we did not conduct additional step-wise distribution analyses as suggested, we appreciate the reviewer's recommendation and believe the revised figure and methodological clarification adequately address the concern regarding data reliability.

Once again, we are grateful for the reviewer's careful reading and constructive feedback, which helped us improve the clarity and rigor of our presentation.

(Figure3)

- I think one of my main issues here is that the manuscript repeatedly uses terms like "stability", "control", or "strategy", yet these are never quantified or measured.

Response: We sincerely appreciate the reviewer's careful attention to the conceptual precision of our manuscript. We fully agree that terms such as "stability," "control," and "strategy" should be used with caution, especially when they are not directly quantified or measured.

In response to this comment, we have conducted a thorough revision across multiple sections of the manuscript—including the Abstract, Introduction, Discussion, Limitation, and Conclusion—to address this issue. Specifically, we have:

- Replaced or rephrased terms like "stability," "control," and "strategy" with more descriptive and kinematically grounded expressions (e.g., "momentum regulation," "kinematic adjustments," or "movement adaptations").
- Where such terms are retained, we have added qualifiers such as "may," "potentially," or "hypothetically," and clearly stated that these interpretations are not based on direct measurements.
- Revised the Limitation section to explicitly acknowledge that conclusions regarding stability or control are speculative and based on observed movement patterns alone.

We believe these revisions improve the conceptual clarity of the manuscript and align the interpretations more closely with the data actually presented. We thank the reviewer again for highlighting this important concern.

"The movement of the center of mass (CoM) in the horizontal plane during the double-support phase is considered critical for maintaining gait performance, but how specific CoM deceleration patterns adapt to these challenges is not fully understood." (Abstract, Page 1)

"The increased the most deceleration and its earlier timing, particularly under challenging conditions, may reflect kinematic adaptations related to momentum regulation and step-to-step coordination, rather than direct indicators of neuromuscular control. These findings provide insight into potential mechanisms underlying gait adaptation in healthy individuals." (Abstract, Page 1)

"These task modifications would shift the timing of this maximum deceleration (MDt) to occur earlier within the gait cycle, potentially indicating anticipatory kinematic adjustments to accommodate altered biomechanical demands." (Introduction, Pages 2, Lines 98-100)

"Although this study focuses exclusively on healthy young individuals, the findings regarding how CoM deceleration patterns are modulated may offer insights into movement adaptations that are relevant to maintaining gait function under varying task constraints. Such knowledge could be

useful in informing the design of interventions aimed at enhancing movement efficiency or safety, for example, in fall prevention or rehabilitation contexts.” (Introduction, Pages 2-3, Lines 102-107)

“These results indicate that alterations in CoM deceleration patterns near toe-off are characteristic of how gait adjusts to changing mechanical demands.” (Discussion, Page 4, Lines 203-204)

“Concurrently, the earlier occurrence of MDt under faster and inclined conditions (Figure 3) may reflect anticipatory kinematic adjustments that help facilitate step-to-step transition. Initiating deceleration earlier relative to the subsequent heel contact could represent a biomechanical adaptation that ensures adequate time and body configuration for weight transfer onto the leading limb, especially under demanding conditions (Hak et al., 2012; Martino et al., 2015). While we did not directly quantify gait stability or neural control, the systematic modulation of MD and MDt observed here could be interpreted as part of a broader pattern of movement adaptation during complex walking tasks.” (Discussion, Page 4, Lines 208-215)

“Therefore, the observed kinematic changes likely result from an interplay between these altered environmental demands and the neuromuscular system’s adaptive responses, rather than providing direct evidence of a specific control mechanism.” (Discussion, Pages 4-5, Lines 217-219)

“The implications of these adjustments for postural control remain speculative, and future work incorporating direct stability-related metrics would be necessary to address this explicitly.” (Discussion, Page 5, Lines 234-236)

“This precise timing may indicate that the peak horizontal deceleration is kinematically coordinated with the termination of the trailing limb’s propulsive phase. Such timing could be biomechanically advantageous for modulating momentum and preparing for swing initiation.” (Discussion, Page 5, Lines 240-242)

“The earlier MDt under challenging conditions, coupled with the increased MD, may reflect an adaptive response aimed at regulating momentum transfer between limbs. For instance, the well-documented increase in ankle plantar flexor activity (soleus and gastrocnemius) during faster or uphill gait (Franz et al., 2012; Neptune et al., 2008) is necessary for propulsion but must be appropriately modulated to manage momentum as the trailing leg unloads. Our findings regarding MD and MDt may represent the kinematic correlates of such adjustments, though direct confirmation would require electromyography (EMG) or kinetic data.

Compared to level gait, where efficient inverted pendulum-like mechanics contribute to minimizing energy cost and supporting step-to-step transitions through effective energy exchange (Donelan et al., 2002; Kuo et al., 2005), the increased the most deceleration of CoM observed during uphill-fast walking in our study—particularly the component directed backward and toward the trailing leg—may reflect a deviation from this economical mechanism. This pattern could represent a less economical but biomechanically necessary movement adjustment aimed at braking momentum under increased mechanical demands, potentially involving increased negative work.” (Discussion, Page 5, Lines 247-260)

“Future investigations should explore whether similar or different patterns of CoM deceleration adjustment are employed by clinical populations (e.g., older adults at risk of falls, individuals post-stroke) and how these patterns may relate to their capacity for gait modulation or response to mechanical challenges. Addressing the limitations of this study—such as incorporating kinetic and EMG measures, examining downhill walking, testing a broader range of speeds and inclines, and including diverse participant groups—will be crucial for developing a more comprehensive understanding and potentially informing clinical applications related to gait assessment, rehabilitation, or fall prevention.” (Discussion, Pages 5, Lines 267-274)

“Studies involving diverse age groups and clinical populations are necessary to determine whether similar patterns of CoM deceleration modulation are utilized and how they might relate to pathology or rehabilitation outcomes.

Fourth, we did not assess muscle activity using EMG. This limits our ability to directly infer the specific neuromuscular strategies employed during the different gait conditions. While our kinematic findings (particularly the modulation of MD and MDt) may suggest associated neuromuscular adaptations, incorporating EMG measurements in future studies would provide crucial insights into the relationship between muscle activation patterns and the observed changes in CoM dynamics.

Fifth, the experiment was conducted on a treadmill without explicit visual feedback manipulations or external balance perturbations. Treadmill walking can differ from overground walking, and the predictable laboratory environment may not fully represent the challenges encountered in daily life. Assessing gait adaptations in more ecologically valid, real-world environments or introducing

controlled perturbations could provide a more comprehensive understanding of how individuals respond to external challenges during locomotion.

Finally, it is important to reiterate that this study relied solely on kinematic data. While CoM kinematics provide valuable information about movement outcomes, directly inferring underlying control mechanisms or quantifying dynamic stability based solely on these measures has inherent limitations. Conclusions regarding ‘stability’ or ‘control’ should be interpreted with caution, as they are inferred from observable movement patterns and not from direct physiological or kinetic evidence.

Addressing these limitations in future research will be essential for building a more robust and physiologically grounded understanding of human gait adaptation across diverse conditions and populations.” (Limitation, Page 6, Lines 298-319)

“These findings highlight that adjustments in CoM deceleration near the time of toe-off are closely associated with how gait adapts to biomechanical demands such as increased speed or incline. The observed kinematic adjustments, particularly the earlier timing of deceleration, may reflect movement adaptations that contribute to maintaining coordination and momentum regulation across steps. However, further research incorporating kinetic and EMG data is needed to confirm the underlying neuromuscular mechanisms.

The interplay between speed and incline influences these horizontal plane dynamics, suggesting that multiple biomechanical factors interact to shape gait responses. While derived from healthy young adults, these insights into fundamental gait adaptation mechanisms may help guide future research exploring how individuals adjust gait in response to mechanical demands and could potentially contribute to developing targeted interventions for individuals with gait impairments.” (Conclusion, Pages 6-7, Lines 327-337)

- The study includes a mixed-sex sample of participants (9 females, 6 males), yet the analysis does not consider sex as a factor or covariate. I therefore recommend that the authors perform an exploratory analysis to determine whether sex significantly influences any of the main outcomes (MD, MDt, SL, or DST). This could involve including sex as a factor in the repeated-measures ANOVA or performing subgroup comparisons. Even if underpowered for formal inference, such an analysis would be valuable for transparency and may help interpret the variance observed across participants. If no meaningful differences are found, please explicitly state that.

Response: We sincerely thank the reviewer for this thoughtful suggestion. In response, we conducted an exploratory subgroup analysis to examine potential sex-related differences in the main outcome variables (MD, MDt, SL, and DST). Independent-samples t-tests were performed for each variable under all four experimental conditions.

The results indicated no statistically significant differences between female and male participants for any of the variables across all conditions ($p > 0.10$ in all cases). However, as the reviewer rightly noted, the sample size within each subgroup was limited ($n = 9$ females, $n = 6$ males), and we acknowledge that the statistical power of this analysis is low.

For transparency, we have added a brief summary of this analysis in the Results section and have included the full details in an appendix table (Appendix table). We believe this addresses the reviewer's concern while maintaining clarity in the main text.

		MD (m/s ²)				MDt (%)			
		Level-moderate	level-fast	uphill-moderate	uphill-fast	Level-moderate	level-fast	uphill-moderate	uphill-fast
Male	Mean	5.28	5.23	3.33	3.43	63.69	63.61	62.78	63.31
	SD	0.93	0.63	0.60	0.82	1.78	1.30	1.75	2.46
Female	Mean	4.60	4.60	3.10	3.17	65.49	65.07	64.47	65.45
	SD	0.79	0.48	0.62	0.53	0.83	1.49	1.50	1.18
ttest	p value	0.26	0.09	0.56	0.59	0.11	0.09	0.14	0.16

		DST (%gait cycle)				SL (mm)			
		Level-moderate	level-fast	uphill-moderate	uphill-fast	Level-moderate	level-fast	uphill-moderate	uphill-fast
Male	Mean	7.97	7.81	23.66	6.07	463.73	528.26	514.98	559.74
	SD	2.15	1.03	38.45	3.55	27.54	31.52	85.42	88.13
Female	Mean	8.06	6.91	7.21	-4.58	455.90	501.97	481.07	528.32
	SD	1.88	1.99	0.99	30.69	28.88	19.62	33.25	36.57
ttest	p value	0.94	0.30	0.38	0.36	0.66	0.18	0.48	0.53

(Appendix table)

“To explore potential sex-related effects, independent-samples t-tests were conducted to compare female and male participants for each of the primary outcome variables (MD, MDt, SL, and DST) under all four gait conditions. While not sufficiently powered for formal inference, this exploratory analysis was included for transparency.” (Methods, Pages 3-4, Lines 160-163)
“The exploratory subgroup analysis revealed no significant sex differences in MD, MDt, SL, or DST under any of the four conditions ($p = 0.09-0.93$).” (Results, Page 4, Lines 173-175)

- Please better explain the method to derive the magnitude of CoM deceleration (MD). The authors mention that they used "most negative value of acceleration magnitude". First "magnitude" typically refers to a positive scalar. Second, are they reporting the most negative value (i.e., highest deceleration) or the maximum of absolute deceleration? This must be clarified.

Response: We thank the reviewer for this important clarification request. We fully agree that the use of the term “magnitude” was misleading in this context, as it typically refers to a positive scalar, whereas our analysis focused on identifying the most negative value of horizontal-plane acceleration (i.e., the greatest deceleration).

To resolve this ambiguity, we have removed the term “magnitude” from the manuscript entirely. We now consistently refer to “the most deceleration (MD),” which is clearly defined as the most negative value of CoM acceleration in the horizontal plane during the double-support phase. This value corresponds to the point of greatest deceleration directed opposite to the direction of movement.

We have revised the relevant sections in the Abstract, Introduction, Methods, and Figure/Table captions to reflect this clarification. The method for calculating MD has also been rewritten in the Methods section to eliminate any potential confusion.

We believe this revision improves the clarity and accuracy of the manuscript, and we thank the reviewer again for bringing this issue to our attention.

“Gait Speed and Incline Modulate Peak Deceleration and Timing of Horizontal Center of Mass Deceleration During Double Support” (Title)

“This study examined how gait speed and incline affect the most deceleration (MD) and its timing (MDt) of center of mass (CoM) movement in the horizontal plane during the double-support phase of gait in healthy individuals.” (Abstract, Page 1)

“Healthy young adults adapt to increased gait speed and incline by modulating both the most deceleration and its timing of CoM movement in the horizontal plane during double support. The increased the most deceleration and its earlier timing, particularly under challenging conditions, may reflect kinematic adaptations related to momentum regulation and step-to-step coordination, rather than direct indicators of neuromuscular control.” (Abstract, Page 1)

“While these foundational studies have provided valuable insights into overall CoM mechanics, transition work, and within-step energy transformations, they did not specifically investigate or quantify the most deceleration and its timing of CoM movement in the horizontal plane, particularly within the double-support phase, as key kinematic variables reflecting momentum management and adaptation to varying speed and incline. We contend that the maximum deceleration of the CoM in the horizontal plane and its timing within the gait cycle, particularly during the double support phase near trailing leg toe-off, represent critical kinematic variables that reflect how the body manages its momentum transfer between steps. Investigating how these specific parameters adapt to changes in speed and incline is therefore essential for gaining deeper insights into the adaptive neuromuscular strategies governing walking.

Therefore, the primary purpose of this study was to clarify the effects of gait speed and incline on these specific characteristics - the most deceleration and its timing of CoM dynamics in the horizontal plane during the double-support phase in healthy young adults walking on a treadmill. We hypothesized the following: (1) Both increased gait speed and the addition of an incline would increase the most deceleration (MD) and its timing (MDt) of center of mass (CoM) movement in the horizontal plane during the double-support phase. (2) These task modifications would shift MDt to occur earlier within the gait cycle, potentially indicating anticipatory kinematic adjustments to accommodate altered biomechanical demands.

Understanding these kinematic responses during healthy gait provides an essential baseline for comparison and may ultimately inform our understanding of gait impairments. Although this study focuses exclusively on healthy young individuals, the findings regarding how CoM deceleration patterns are modulated may offer insights into movement adaptations that are relevant to maintaining gait function under varying task constraints. Such knowledge could be useful in

informing the design of interventions aimed at enhancing movement efficiency or safety, for example, in fall prevention or rehabilitation contexts.” (Introduction, Pages 2-3, Lines 83-107)

“The variables used for analysis were the double support time ratio (DST), step length (SL), the most deceleration of CoM in the horizontal plane (MD), and its timing (MDt). MD was calculated as follows. The instantaneous CoM velocity vector in the horizontal plane (combining anteroposterior and mediolateral components) was calculated, and horizontal-plane acceleration was obtained by numerical differentiation. The most negative value of this acceleration during the double-support phase was extracted as MD, representing the largest deceleration acting opposite to the direction of movement.” (Methods, Page 3, Lines 149-155)

“Our primary findings supported our hypotheses: both faster speeds and uphill walking significantly increased MD, and these conditions shifted the MDt earlier in the gait cycle compared to moderate-speed level walking. These results indicate that alterations in CoM deceleration patterns near toe-off are characteristic of how gait adjusts to changing mechanical demands.” (Discussion, Page 4, Lines 200-204)

“Compared to level gait, where efficient inverted pendulum-like mechanics contribute to minimizing energy cost and supporting step-to-step transitions through effective energy exchange (Donelan et al., 2002; Kuo et al., 2005), the increased the most deceleration of CoM observed during uphill-fast walking in our study—particularly the component directed backward and toward the trailing leg—may reflect a deviation from this economical mechanism. This pattern could represent a less economical but biomechanically necessary movement adjustment aimed at braking momentum under increased mechanical demands, potentially involving increased negative work. In summary, this study demonstrates that healthy young adults systematically modulate the most deceleration and its timing of peak CoM in the horizontal plane during double support when adapting to increased walking speed and incline.” (Discussion, Page 5, Lines 254-263)

“Specifically, we found that both faster walking speeds and uphill conditions led to an increase the most deceleration and shifted its timing to occur earlier within the gait cycle.” (Conclusion, Page 6, Lines 324-326)

- **“...a second-order Butterworth low-pass filter at 10 Hz” could you justify this?**

Response: Thank you for pointing this out. We have now added a detailed justification for our filtering choice in the Methods section. Specifically, we referenced the validation study by Koltermann et al. (2018), which systematically compared various filter types, orders, and cutoff frequencies in the context of biomechanical signals. Their findings showed that the second-order Butterworth filter provides a stable and accurate filtering performance across a range of dynamic measurements, with a 10-13 Hz cutoff frequency offering reliable results for human movement data, including center of pressure and potentially center of mass trajectories. Therefore, our selection of a 10 Hz cutoff was based on this empirical evidence, ensuring that relevant signal components are retained while noise is effectively attenuated.

“Marker coordinates were processed with a second-order Butterworth low-pass filter at 10 Hz, based on previous validation studies that demonstrated this filter’s effectiveness in minimizing high-frequency noise while preserving the biomechanical signal of interest during dynamic tasks (Koltermann et al., 2018). This combination was shown to provide an optimal balance between signal fidelity and noise reduction, particularly in human movement analysis.” (Methods, Page 3, Lines 142-146)

[Reference] Koltermann, J. J., Gerber, M., Beck, H., & Beck, M. (2018). Validation of Various Filters and Sampling Parameters for a COP Analysis. *Technologies*, 6(2), 56.
<https://doi.org/10.3390/technologies6020056>

- **My last comment is about the speed conditions (0.83 m/s and 1.0 m/s). These are relatively close together, and both fall within a relatively narrow functional range for healthy young adults. Isn't it? The limited speed range may constrain the generalisability of the findings and the ability to meaningfully interpret the reported effects. I suggest the authors discuss this limitation.**

Response: We thank the reviewer for this insightful comment. We agree that the walking speeds used in our study (0.83 m/s and 1.0 m/s) fall within a relatively narrow functional range and are somewhat lower than the typical preferred walking speeds of healthy young adults. This may limit the generalizability of our findings to more natural walking conditions. In response to this valuable suggestion, we have revised the Limitation section to explicitly acknowledge this issue. Specifically,

we now note that the limited speed range may constrain the interpretation of the reported effects and highlight the need for future studies to explore a broader range of walking speeds.

“Second, our investigation focused on a limited range of gait speeds (moderate and fast) and a single uphill incline (+6°). Specifically, the walking speeds of 0.83 m/s and 1.0 m/s were chosen to ensure consistency across participants. However, these speeds fall below the typical preferred walking speeds observed in healthy young adult males, and both lie within a relatively narrow functional range. As such, the limited variation in gait speed may constrain the generalizability of our findings and reduce the ability to fully interpret the effects of speed on CoM dynamics. This narrow scope may also not capture the full spectrum of gait adaptations that occur at slower speeds, steeper inclines, or during downhill walking. Future research should explore a broader range of walking speeds and inclines to better reflect natural walking conditions and provide a more comprehensive understanding of how these variables interactively influence CoM dynamics during double support.” (Limitation, Page 6, Lines 285-294)

Second decision letter

MS ID#: bio.062037R1

MS TITLE: Gait Speed and Incline Modulate Peak Deceleration and Timing of Horizontal Center of Mass Deceleration During Double Support

AUTHORS: Shizuku Terui, Nanami Kanda and Keisuke Hirata

I've had a chance already to fully read through your rebuttal along with your manuscript edits, and I am happy to tell you that your manuscript has been accepted for publication in Biology Open, pending our standard publication integrity checks. It was accepted on 04 Jun 2025.